# Approach of Acromegaly during Pregnancy

**DOI:** 10.3390/diagnostics12112669

**Published:** 2022-11-02

**Authors:** Alexandru Dan Popescu, Mara Carsote, Ana Valea, Andreea Gabriela Nicola, Ionela Teodora Dascălu, Tiberiu Tircă, Jaqueline Abdul-Razzak, Mihaela Jana Țuculină

**Affiliations:** 1Department of Endodontics, Faculty of Dental Medicine, University of Medicine and Pharmacy of Craiova, 200349 Craiova, Romania; 2Department of Endocrinology, Carol Davila University of Medicine and Pharmacy & C.I. Parhon National Institute of Endocrinology, 011683 Bucharest, Romania; 3Department of Endocrinology, Iuliu Hatieganu University of Medicine and Pharmacy & Clinical County Hospital, 400012 Cluj-Napoca, Romania; 4Department of Oro-Dental Prevention, Faculty of Dental Medicine, University of Medicine and Pharmacy of Craiova, 200349 Craiova, Romania; 5Department of Orthodontics, Faculty of Dental Medicine, University of Medicine and Pharmacy of Craiova, 200349 Craiova, Romania; 6Department of Infant Care–Pediatrics–Neonatology, Romania & Doctoral School, University of Medicine and Pharmacy of Craiova, 200349 Craiova, Romania

**Keywords:** acromegaly, pregnancy, growth hormone, prolactin, somatostatin, octreotide, lanreotide, diabetes mellitus, somatotropinoma, pituitary, cabergoline, pegvisomant

## Abstract

Acromegaly-related sub/infertility, tidily related to suboptimal disease control (1/2 of cases), correlates with hyperprolactinemia (1/3 of patients), hypogonadotropic hypogonadism—mostly affecting the pituitary axis in hypopituitarism (10–80%), and negative effects of glucose profile (GP) anomalies (10–70%); thus, pregnancy is an exceptional event. Placental GH (Growth Hormone) increases from weeks 5–15 with a peak at week 37, stimulating liver IGF1 and inhibiting pituitary GH secreted by normal hypophysis, not by somatotropinoma. However, estrogens induce a GH resistance status, protecting the fetus form GH excess; thus a full-term, healthy pregnancy may be possible. This is a narrative review of acromegaly that approaches cardio-metabolic features (CMFs), somatotropinoma expansion (STE), management adjustment (MNA) and maternal-fetal outcomes (MFOs) during pregnancy. Based on our method (original, in extenso, English—published articles on PubMed, between January 2012 and September 2022), we identified 24 original papers—13 studies (3 to 141 acromegalic pregnancies per study), and 11 single cases reports (a total of 344 pregnancies and an additional prior unpublished report). With respect to maternal acromegaly, pregnancies are spontaneous or due to therapy for infertility (clomiphene, gonadotropins or GnRH) and, lately, assisted reproduction techniques (ARTs); there are no consistent data on pregnancies with paternal acromegaly. CMFs are the most important complications (7.7–50%), especially concerning worsening of HBP (including pre/eclampsia) and GP anomalies, including gestational diabetes mellitus (DM); the best predictor is the level of disease control at conception (IGF1), and, probably, family history of 2DM, and body mass index. STE occurs rarely (a rate of 0 to 9%); some of it symptoms are headache and visual field anomalies; it is treated with somatostatin analogues (SSAs) or alternatively dopamine agonists (DAs); lately, second trimester selective hypophysectomy has been used less, since pharmaco-therapy (PT) has proven safe. MNA: PT that, theoretically, needs to be stopped before conception—continued if there was STE or an inoperable tumor (no clear period of exposure, preferably, only first trimester). Most data are on octreotide > lanreotide, followed by DAs and pegvisomant, and there are none on pasireotide. Further follow-up is required: a prompt postpartum re-assessment of the mother’s disease; we only have a few data confirming the safety of SSAs during lactation and long-term normal growth and developmental of the newborn (a maximum of 15 years). MFO seem similar between PT + ve and PT − ve, regardless of PT duration; the additional risk is actually due to CMF. One study showed a 2-year median between hypophysectomy and pregnancy. Conclusion: Close surveillance of disease burden is required, particularly, concerning CMF; a personalized approach is useful; the level of statistical evidence is expected to expand due to recent progress in MNA and ART.

## 1. Introduction

Acromegaly, a disease induced by a pituitary GH (Growth Hormone) producing tumors in 95% of cases, is part of a larger category of endocrine entities associated with hypophyseal tumors with an increasing ratio in the general population (incidence and prevalence of 3.9–7.4, respective 76–116 cases per 100,000 per year, respectively, particularly for acromegaly, with values of 0.2–1.1 and 2.8–13.7, respectively) [1,2]. The mean age at which acromegaly is diagnosed is within the fifth decade of life, with an earlier detection with a 4–5-year median; while modern approaches may even detect completely asymptomatic cases, for instance, as a pituitary incidentaloma as part of silent GH pituitary neuroendocrine tumors [2,3,4,5].

Chronic excess of GH and IGF1 (Insulin-like Growth Factor 1) has dramatic consequences concerning cardio-metabolic parameters (high blood pressure, glucose and lipid profile anomalies, left ventricular hypertrophy, diastolic, systolic and endothelial dysfunction, and even heart failure), cosmetic features (course facial and acral enlargement), respiratory aspects (sleep apnea), muscular-skeletal issues (arthropathy, osteoporosis and fragility fractures, especially vertebral fractures) and a selective oncologic risk (colonic cancer), causing an increased morbidity and mortality with a massive impairment of quality of life [3,6,7,8,9,10,11,12,13,14].

A coordinated team of different specialists is required to treat the associated complications in addition to the targeted approach of the pituitary tumor; this mostly involves selective trans-sphenoidal hypohysectomy as the first line of therapy to address somatotropinoma expansion and GH-IGF1 excess and/or pharmaco-treatment underlying two generations of somatostatin analogues (SSAs)—octreotide, lanreotide, respective pasireotide, dopamine analogues (DAs)—bromocriptine and cabergoline, and GH receptors antagonist—pegvisomant, as well as radiotherapy, especially gamma knife type, in selected cases [15,16,17,18,19,20,21,22]. These aspects are detailed in numerous guidelines from different endocrine, surgery and multidisciplinary societies [15,16,17,18,19,23]. General increased life expectancy and advanced progress of acromegaly management associate not only a higher number of patients with somatotropinomas among an elderly population and a younger age group at first detection, but also a larger number of women of reproductive age that are potentially able to carry a child and deliver a healthy new-born [24,25,26,27,28,29].

### Aim

We aim to focus on cardio-metabolic features, somatotropinoma evolution, maternal and fetal outcomes, and management adjustment regarding pregnancy in acromegalic females.

## 2. Methods

This is a narrative review. We revised the papers addressing the mentioned purpose from January 2012 to September 2022 (in extenso, English—published articles on PubMed). We included original studies with different levels of statistical evidence (from original studies to cases reports) following the combination of keywords “acromegaly” (or “pituitary tumor”, “somatostatin analogue”, “octreotide”, “lanreotide”, “pegvisomant”, “growth hormone”, “IGF1”) and “pregnancy” (alternatively, “lactation”).

## 3. GH-IGF1 Axes during Pregnancy: Non-Acromegalic and Acromegalic Females

Generally, GH excess is responsible for multiple complications concerning acromegaly by inducing a pro-diabetogenic status, microvascular damage (like retinopathy) and insulin resistance, as well as lipolytic and pro-anabolic effects [30,31,32].

Physiological pregnancies are associated with massive changes of hormone status throughout the three trimesters of gestation, including those already known to underlie different pathogenic loops in acromegaly, such as GH, IGF1, insulin and estrogens. A failure of beta-pancreatic cells to adjust their insulin secretion during the first trimester might be a contributor to gestational DM [30]. Placenta-released kisspeptin modulates pancreatic insulin production via its receptor GPR54 [31,32]. The placenta cross-talks with the hypophyseal gland, among others, by releasing placental GH [32,33].

Normal pulsatile secretion of GH (also, named GH1), a hormone that is stimulated by the hypothalamus-released GH-RH (GH-Releasing Hormone) and gastric-released ghrelin, is registered outside pregnancy and it is presented during the first trimester of a physiological pregnancy. Pituitary GH does not cross the placenta; thus, there are no negative effects on the fetus even in acromegalic patients. In females with intact GH function, IGF1 physiologically decreases during the first trimester of pregnancy, while GH remains stable [33,34,35].

Then, the placental GH (also, known as GH2) takes control of pituitary GH under the physiological circumstances of a non-acromegalic gestation [33,34]. Placental GH secretion is continuous, not pulsatile; it is first detected at week 5, and by week 15 it progressively becomes the main circulating form of GH, which stimulates liver production of IGF1 and inhibits pituitary GH in non-acromegalic females [33,34,35]. GH2 remains the main player until week 37 [33,34]. The shift from one type of GH to another might explain the small reduction of IGF1 levels amid the first pregnancy trimester [33,34,35,36,37].

Physiologically, IGF1 will later increase due to placental GH stimulation (a peak during week 37); it provides essential growth-promoting effects on both the placenta and the fetus (among others, IGF1 represents the major regulator of development concerning most organs, including the central nervous system where it controls glucose metabolism) [33,34,35,36,37]. In acromegaly, IGF1 usually remains stable throughout the remaining period of gestation time because of the estrogen-induced state of GH resistance that prevents somatotropinoma-associated production of pituitary GH to have clinical and metabolic fetal consequences [36,38].

However, after delivery, placental GH decreases from the first day, while pituitary GH progressively rises, similar to IGF1 (as expected, with higher levels in acromegalic females than in non-acromegalic females, probably due to GH escape from estrogen blockade) [38,39]. For instance, one study on ten pregnancies from eight acromegalic individuals aged between 24 and 37 years showed that IGF1 levels before and during pregnancy were similar, but were found to be significantly increased postpartum versus pre-partum [39]. GH assays do not detect the hormone with placental origin [40,41].

Opposite to pituitary GH, placental GH, the product of syncytiotrophoblast, is not affected by ghrelin, nor by GH-RH, and hypoglycemia decreases GH2 [38,41,42]. Both GH1 and GH2 are inhibited by glucose and, as mentioned, stimulate the liver’s production of IGF1. Arginine increases GH1, but GH2 response is less predictable [38,41,42]. Moreover, IGF2 displays a modest increase of 20–30% throughout the pregnancy, also playing an important role in pre- and post-labor growth and development [43,44] (Figure 1).

## 4. Sub/Infertility Issues in Acromegaly

Hypopituitarism (including central hypogonadism), reported in 10–80% of acromegalic cases, and hyperprolactinemia (affecting more than one third of the patients) are associated with a higher rate of subfertility/infertility in acromegalic patients when compared to a non-acromegalic population of the same age, in both females and males [45,46,47,48,49,50,51,52,53,54,55]. Pituitary insufficiency and increased prolactin status are related to larger tumors; 90–95% of pituitary GH secreting tumors are macroadenomas; the additional negative role concerning fertility status is caused by secondary DM [46,47,48,49,50,51,52,53,54,55]. For instance, one study from 2022 on 529 acromegalic individuals identified a rate of hyperprolactinemia of 39.1% and a prevalence of hypopituitarism of 34.8%, with the gonadal axis being the most affected pituitary axis; there was a hypogonadism frequency of 29.7% [45].

In males, the same mechanisms are contributors to erectile dysfunction, while GH growth-promoting effects may be involved in benign prostatic hypertrophy [46,47,48].

GH-induced LH (Luteinizing Hormone) hypo-responsiveness represents a proposed mechanism of male infertility in acromegaly [49]. Moreover, some studies showed an impairment of semen quality (but not all studies agree) [56,57,58].

Hypogonadism requires estrogen/progesterone, respective testosterone substitution, while hypogonadotropic hypogonadism in females and males with aggressive pituitary tumors that are partially responsive to standard therapy might be treated with gonadotropins or Gonadotropin Releasing Hormone (GnRH) protocols to increase the fertility potential [50]. Alternatively, the use of clomiphene citrate, a selective estrogen receptor modulator (SERM), stimulates LH and FSH (Follicle Stimulating Hormone) with good results on fertility rates; the drug was proposed as an add-on therapy added to acromegaly accompanied by hypogonadism which is not controlled by SSAs and/or DAs [59,60,61]. However, despite well-known fertility issues in acromegalic patients, spontaneous pregnancy has been reported [51]. Of note, the control of tumor volume and hormone excess might reverse some of the underlying mechanisms in acromegaly-related infertility, which is closely related to recovery of pituitary function and improvement of overall disease burden [62,63].

## 5. Cardio-Metabolic Features in Pregnant Females with Acromegaly

We already know that an important ratio of acromegalic patients display uncontrolled or inadequately controlled disease despite pituitary surgery and/or medical treatment and even radiotherapy. Several risk factors are independent predictors of a more aggressive behavior from the first diagnostic or of a poor response to standard therapy, for instance, large, invasive tumors have a lower rate of remission after hypophysectomy; a partial response to first line SSAs (octreotide and lanreotide) is expected in cases with sparsely granulated adenomas, lack of somatostatin receptor type 2 (SSTR2) at immunohistochemistry report, very young patients, particular imaging aspects such as T2-hyperintensity, and mutations of the AIP gene (aryl hydrocarbon receptor-interacting protein) [64,65].

One the other hand, active disease under first-generation SSAs represents an indicator of a more severe form, most patients becoming candidates for either pasireotide or pegvisomant in terms of pharmacological intervention [20,66,67]. Other comorbidities with a major impact on overall health are represented by cardio-metabolic complications, particularly HBP and DM, which are partially controlled by standard intervention for acromegaly [68,69,70]. With respect to the complex management of acromegaly nowadays that, yet, is associated with almost half of the cases with suboptimal control (depending on criteria), pregnancy in females with somatotropinoma remains an exceptional event. However, over the years, the approach has been refined due to a larger number of published cases or studies which are almost exclusively of observational type. We identified the first three publications on PubMed concerning this particular topic from 1949, respective 1954 [71,72,73,74].

Pregnancy in acromegalic females, an unusual aspect, is reported spontaneously despite infertility concerns, as mentioned earlier, or due to the use of ovulation inductors, GnRH or gonadotropines and, lately, via assisted reproductive techniques [50,51,59,60,75]. Interestingly, the acromegaly in these women was either diagnosed previously and treated with one, two or all three lines of management (including radiotherapy) or it was diagnosed for the first time during gestation, which challenges the biochemistry assessments due to placental GH overlap [40,41].

Overall, pregnancy in acromegalic females involves the following issues: cardio-metabolic features which were found to be the main players during gestation among the heterogeneous panel of disease-related comorbidities; potential tumor expansion; particular materno-fetal outcomes; changes in the standard management for acromegaly and specific iatrogenic/teratogenic concerns (if any).

Despite the theoretical rationale of focusing on a functional pituitary adenoma syndrome, the evolution of arterial hypertention/pre-eclampsia/eclampsia and the glucose profile are the real issues regarding pregnant acromegalic women rather than somatotropinoma growth (which is possible, too) [39,76,77,78]. The best predictor of these cardio-metabolic features remains the degree of disease control at the moment of conception, which is routinely reflected in the blood level of IGF1, regardless of whether the subject is under medical treatment at the moment of pregnancy confirmation [76,77,78].

One retrospective study identified 13 pregnancies in acromegalic females; the most frequent complications were gestational DM and HBP (7.7%), while no tumor expansion was registered during the gestation period. This subgroup is part of a larger cohort of 47 acromegalic women of reproductive age, among whom 38% had normal gonadal function, 38% had gonadal dysfunction without the central component of the hypogonadism and 24% had hypogonadotropic hypogonadism. The females from the second group had higher prolactin levels and a longer estimated duration of disease until its recognition. A total of 64% of women had low AMH (anti-Mullerian hormone) in 14 females that were younger than 45 years old [76]. Gonadal dysfunction was remitted in 66% of cases, suggesting the potential of reversible endocrine profile, as previously mentioned [62,63,76].

In another single-center study, 11 females (*N* = 14 pregnancies) who were 34 ± 3.6 years old were analyzed concerning glucose profile during pregnancy. The patients were either treated with SSAs following hypophysectomy (N1 = 6) or as a first line therapy (N2 = 5). SSAs were stopped at the moment of pregnancy confirmation, except for one female who received for the first time the diagnostic of acromegaly during gestation and she was further exposed to SSAs until weeks 12–18. Half of the patients (*N* = 7) had gestational DM (one female even had it twice throughout both her pregnancies), which was confirmed either on fasting blood glucose or oral glucose tolerance test. Pre-conception IGF-1 was suboptimal in four cases with DM and two without gestational DM. Women who developed DM had a higher body mass index before conception, positive family for type 2 DM, but not a more advanced age. Screening for gestational DM in acromegaly is mandatory (including oral glucose tolerance test) [77].

Moreover, we mention a retrospective study on 31 pregnancies (*N* = 20 women with somatotropinoma); out of these, 4 females had an abortion. The cardio-metabolic profile confirmed the worsening of HBP in 45% of cases, and of glucose control in 32% of the 27 pregnancies, these representing the most frequent complications associated with the gestation period. The approach of acromegaly amid gestation included different aspects such as: 3/27 women had selective hypophysectomies; IGF1 normalization occurred in 23/27 cases; 15/27 females were withdrawn from medical treatment. No maternal-fetal death was registered. Fetal outcomes included two congenital malformations and one case of macrosomia [78].

Overall, DM and HBP represent major clinical elements that should be given particular attention during pregnancy in an acromegalic subject, including newly detected cases. As far as we currently know, cardio-metabolic complications do not increase the risk of materno-fetal outcomes more than generally found in non-acromegalic populations [79,80,81,82]. We do not have enough data to sustain a different evolution and management during pregnancy with respect to gestational DM overt previously (secondary) DM/pre-DM which is generally found in 10–70% of all cases, neither between prior known, secondary HBP (which is expected to worsen) overt gestational HBP; the decision of intervention should be made from an individual perspective, but close surveillance is required and periodic checkups of glucose profiles are mandatory [83,84]. The risk of gestational diabetes is 15% with regard to the general population; there is a higher risk of fetal outcomes such as: macrosomia, stillbirth, large for gestational age (or small), lower APGAR, respiratory distress syndrome, neonatal hypoglycemia and jaundice, respective of maternal outcomes: preterm birth, need of cesarean section, or developing type 2 DM later in life [33,85,86,87]. The pathogenesis involves beta pancreatic cell dysfunction (anomalies of insulin secretion) and insulin resistance, but also a dysregulation of the IGF system, mostly higher levels of IGF1 and IGF2, and reduced IGFBP1 and IGFBP4 (IGF Binding Protein); IGF1 and IGF2 are essentially important for glucose transport to the fetus [33,85,86].

IGF1, as a surrogate marker of active acromegaly and, potentially, the presence of other risk factors (family history of DM or high body mass index), provides special clues for developing cardio-metabolic gestational complications [77].

## 6. Somatotropinoma Evolution during Gestation

Tumor expansion might be found throughout pregnancy, particularly in macroadenomas that usually represent 90–95% of all somatotropinomas; we do not have enough evidence to conclude a clear rate of tumor growth during this period of time, varying from 0% to 9% [76,88]. Neurosurgery is feasible within the second trimester; most recently, selective pituitary resection has been less preferred during pregnancy, being restricted to uncontrolled cases under medical therapy which is associated with a larger body of evidence concerning its safety [27,38,78]. Particular clinical clues of tumor expansion are headache and visual field damage (which should be checked every 4–6 weeks in macroadenomas) [88,89]. Initially, somatrotropinoma increase might be approached via pharmacological intervention in most cases. Additionally, magnetic resonance imaging provides enough information on tumor growth, but it is not routinely recommended for cases with good clinical outcome in pregnancy; however, a postpartum imaging assessment is essential; intra-pregnancy use of intravenous contrast as gadolinium should be carefully assessed based on a personalized decision [90,91,92].

As an example, we mention a case report of a 30-year-old female presented within the second trimester with progressive right eye vision loss. She was confirmed with acromegaly and underwent trans-nasal trans-sphenoidal hypophysectomy (week 22) with visual improvement and she underwent cabergoline therapy (1 mg/week) until she delivered by cesarean a full-term, healthy newborn [93].

The use of SSAs or DAs may postpone the surgery after delivery; a lack of medical therapy and the natural evolution of an aggressive somatotropinoma during gestation sometimes requires prompt postpartum hypophysectomy [94].

## 7. Specific Medical Therapy for Acromegaly during Pregnancy

Current data recommend that SSAs, DAs or pegvisomant should be stopped at the moment of pregnancy confirmation, ideally before conception, since biochemical escape is unlikely. However, gestation exposure to pharmacological intervention does not represent an indication of pregnancy termination, according to current information, but a personalized, multidisciplinary decision is mandatory. Most studies include the first trimester as exposure time (if any). Recent evidence concerning patients who were under medical therapy for a longer period of time during gestation months showed a good safety profile [95]. Particular aspects concerning surveillance of glucose profile are added by the use of SSAs to cardio-metabolic features in acromegalic pregnancies [95,96,97,98,99].

One study from 2020 on 127 women (*N* = 141 pregnancies) followed particular aspects regarding first generation SSA exposure, with women treated during pregnancy (N1 = 67 pregnancies in 62 females) or not treated (N2 = 74 pregnancies in 65 individuals). Materno-fetal outcomes were similar between N1 and N2 (meaning gestational DM, HBP, headache, delivery, and birth term, height and weight of the new born). One case (1/141) had ureteral stenosis (from N1) [95]. The optimum time frame of SSA exposure, if any, is also an open question; this study also addressed this issue: from the N1 group, 36 pregnancies were associated with SSA therapy only throughout the first trimester, while 20 females had therapy for more than the first trimester; both subgroups had similar materno-fetal outcomes [95]. Of note, this study on 141 acromegalic pregnancies represents the largest original study on this particular topic, according to our method of research.

Based on the data we currently have, SSAs should be stopped at the moment of pregnancy confirmation, but according to a case-by-case decision, the treatment may be prolonged probably throughout the entire first trimester or even longer in exceptional cases. The main indication of SSA gestational exposure is probably the amelioration of headache to improve the quality of life and to reduce the use of analgesics which is in any case restricted during pregnancy [95,96,97]. Headache is related to potential tumor growth which turned out not to be dramatic in most of the published cases. However, exceptions are found in large somatotropinomas that are at higher risk of displaying local mass effects even with dramatic visual impairment. We mention one 32-year-old female who was first confirmed with acromegaly during week 11 of gestation; she consecutively developed bitemporal hemianopia (week 20) and, while she refused pituitary surgery, daily subcutaneous octreotide 100 µg was offered and increased to 150 µg in order to control the visual damage. She underwent selective cesarean at week 34 after, also, developing gestational DM (which required daily insulin therapy). The newborn was healthy and remained so for the next 2 years of follow-up [96]. During the gestation period, tumor expansion may be managed only with medical therapy, especially octreotide, as indicated by most data, unless hypophysectomy is provided in severe cases [77,95,96,97].

Another scenario involves inoperable giant somatotropinomas (with a diameter larger than 4 cm) in patients who, through achieving biochemical control under medical treatment, might conceive [97]. For instance, we mention a young female with macroadenoma severely invasive with respect to the cavernous sinus, the right orbital cavity, the sphenoid sinus, etc., which was rapidly responsive to ocreotide LAR in terms of controlling the headache and bilateral hemianopsia and allowing a good fertility profile in order to get pregnant. The pharmaco-therapy was stopped during gestation while she experienced no adverse outcome and delivered a healthy newborn. Post-partum imaging assessment proved no remarkable tumor evolution [97].

With respect to the three types of medical therapy that may be offered to an acromegalic individual (SSAs, DAs and pegvisomant), most studies introduce first-generation SSAs (with the exception of what we already know concerning the use of cabergoline and bromocriptine in pregnant women with micro- or macro-prolactinomas) [100,101,102].

Concerning the first-generation SSA lanreotide, we identified less data when compared to octreotide [103]. No report of the second-generation SSA pasireotide was identified amid gestation. The drug was introduced into daily practice later than first-generation SSAs targeting tumors with a higher expression of SSTR5 rather than type 2; thus, it found its way in acromegaly management considering that 24–65% of acromegaly cases might not achieve optimal control under first-generation SSAs [104,105,106,107]. Not all tumors display the same configuration SSTS, which may indicate SSA response; the immunohistochemistry report after selective hypofisectomy provides a useful assessment of the SSTR2/5 ratio [108].

A limited amount of data are focused on pegvisomant and, so far, it seems to have a good safety profile, too; generally, the drug represents a second line of medical therapy alternative to the modern introduction of pasireotide [109]. A summary paper from 2014 introduced the expansion of ACROSTUDY that identified before 2014 a total of 35 pregnancies related to pegvisomant exposure at the moment of conception, meaning 27 females and 8 acromegalic men conceiving with their non-acromegalic partners. Two women (2/27) carried two pregnancies each; the drug dose at the moment of conception was 15.3 mg/day (between 4.3 mg and 30 mg/day); three females continued the treatment throughout the entire gestation period. The data were obtained for ten pregnant acromegalic females who delivered full-term healthy newborns. Another five females had an elective pregnancy termination without fetal anomalies, another two had spontaneous abortions apparently unrelated to the GH receptor antagonist, while another one had an ectopic pregnancy. The drug seemed safe, although a recommendation of its withdrawn during pregnancy is done [110].

A paper for 2020 introduced case series of four pregnancies in three females treated with this drug: in one case, therapy was stopped three days before embryo transfer for two pregnancies (both with successful outcomes for each newborn); the other two cases had pegvisomant withdrawn at the moment of pregnancy confirmation. No fetal complications were registered (one of the females had a twin pregnancy) [111].

Preconception exposure to pegvisomant seems safe from a materno-fetal outcome point of view. Ideally, therapy should be stopped before conception or assisted reproductive procedures, but, alternatively, the medication may be withdrawn at the moment of pregnancy confirmation as well. Pregnancy in an acromegalic female treated with this drug might be associated with the mentioned cardio-metabolic complications independently of pegvisomant use, as single medical therapy or in combination with SSAs/DAs [76,77,111].

Interestingly, one case of a pregnant woman with first-generation SSAs for non-pituitary neuroendocrine tumor showed a stable disease and a good safety profile for this medication (despite low level of statistical evidence we currently have) [112].

The use of ocreotide for familial hyperinsulinemic hypoglycemia has been suggested, but some authors reported that that might induce fetal growth restriction. However, this aspect is not confirmed by most data we have on acromegaly [113,114].

## 8. Prolactin, DAs and Pregnancy

A total of 10% of somatotropinomas might co-secrete prolactin; thus, DAs, meaning cabergoline and bromocriptine, represent a useful second line of medical therapy (after SSAs) or even as a first option for medical treatment in cases with a mild elevation of IGF1 that do not require SSAs (for instance, mild residual disease after pituitary surgery was already performed) [115,116].

Among functional and non-functional pituitary neuroendocrine tumors, prolactinomas are the subject of most interference with pregnancy due to the prolactin role in gonadal function. DAs should be stopped at gestation confirmation; the risk of tumor growth is irrelevant for microprolactinomas, but some macroprolactinomas might expand; thus, DAs should be reinitiated [117,118,119]. We mention a study introducing DA, not SSA, therapy during gestation in acromegalic patients. This is part of the Irish Pituitary Study. A total of 17 pregnancies in 12 acromegalic females were followed. The assessments before conception showed: 11/12 women had macro-somatotropinomas and 1/12 had a microadenoma; 5/17 pregnancies achieved optimal acromegaly control; 6/17 pregnancies had DA exposure among pregnancy; the other medical treatments specifically addressing the disease were stopped. Maternal outcomes included: no tumor expansion and associated signs or symptoms including visual field anomalies; no events concerning HBP or gestational DM. Fetal outcomes involved: 15/17 healthy newborns at term; 1/17 pre-eclampsia-related emergency cesarean (week 32); and 1/17 elective C-section for twin pregnancy (week 35) [118]. The study represents one more confirmation of a safe profile with respect to pregnancy outcome in acromegalic individuals. However, other studies identified a small percent of congenital malformations [78,95].

We identified a retrospective study that introduced females with different types of pituitary tumors amid gestation (out of 113 patients, there were 83 prolactinomas and 21 acromegaly cases). The overall rate of tumor expansion during pregnancy was low and there was no significant risk of hypopituitarism; prolactiomas, especially those with uncontrolled disease before conception, were at higher risk of tumor volume increase when compared to other types of hypophyseal adenomas; the outcomes were similar between females under medical therapy and those not. Karaca Z et al. also identified the following malformations: neural tube defect and microcephaly associated with cabergoline exposure, Down syndrome and corpus callosum agenesis related to bromocriptine use, unilateral congenital cataract, craniosynostosis and microcephaly in acromegalic females [120]. One prospective, population-based cohort from the UK involved 71 females with pituitary tumors (including three cases with acromegaly and 49 macroprolactinomas) and no pregnancy outcomes were registered (meaning HBP and preeclampsia, preterm birth and stillbirth) [121].

## 9. Pregnancy Outcome and Materno-Fetal Complications

As far as we currently know, despite a low level of statistical evidence, but with a growing number of publications lately, pregnancy seems safe regarding materno-fetal outcomes in most of the reported cases/studies. However, close multidisciplinary surveillance is required, as well as preconception counselling [75,122,123,124].

As previously mentioned, gestational DM correlates with a higher risk of different materno-fetal outcomes in the general population [85,86,87]. Specific reports on the acromegalic population include isolated cases of macrosomia, ureteral stenosis, unilateral congenital cataract, craniosynostosis and microcephaly [78,95,120]. The degree of acromegaly control does not seem associated with the risk of fetal malformations [75].

One retrospective study from 2021 conducted by Das L et al. on 12 acromegalic females (a total of *N* = 14 pregnancies) who were followed between 2010 and 2019 introduced two subgroups: one category (N1 = 5) with active acromegaly at the moment of pregnancy confirmation, another category (N2 = 9) included patients with controlled disease versus an acromegalic cohort (N3 = 75). N1 and N2 had macroadenomas which were referred for surgery prior to pregnancy (except for two females who were first recognized with acromegaly during gestation and they were treated with medication during pregnancy and referred to hyphophysectomy afterwards). Pregnancy confirmation followed the surgical procedure for GH producing adenoma within 0.8–5.1 years (a median of 2 years). Techniques of assisted reproduction were necessary in 21.4% of cases. Except for one case with gestational HBP and consecutive preterm baby, N1 and N2 cohorts had term delivery (*N* = 13) and normal APGAR. No congenital malformations were identified. Materno-fetal outcome was similar between N1 and N2, despite the fact that GH-IGF1 levels and somatotropinoma volume were statistically significantly higher in the N1 subgroup. Moreover, postpartum GH and tumor remnants were similar between N1 and N2. The study confirms that the pregnancy outcome/materno-fetal complications are similar to the non-acromegalic population, including active acromegaly [75]. We may further expect a larger number of successful pregnancies in acromegalic females due to the spreading of assisted reproductive procedures [75,122].

## 10. Discussion

The topic of acromegaly and pregnancy highlights several major aspects. First, there is the question of what we should further expect concerning the level of statistical evidence in this particular domain. On one hand, there is still a significant subgroup of patients with inadequate disease control that impairs the fertility rate; thus, pregnancy becomes an exceptional event in acromegalic females of reproductive age. On the other hand, early recognition of GH excess and/or somatotropinoma and access to multi-lined management in some countries already improved the overall prognostic, so young women diagnosed with the condition may get to the point of delivering a healthy newborn. An additional contributor is represented by easier access to assisted reproduction. With regard to this particular concern, a performant prenatal diagnostic might help in acromegaly forms with genetic backup that are expected to present the disease at younger ages when compared to sporadic cases and that are also expected to affect the offspring. For instance, one single case of familial acromegaly is reported to have a pre-natal diagnostic done on a newborn with familial X-linked acrogigantism inherited from the mother (de novo duplication of Xq26.3) [125,126].

Second, a limited amount of data are published concerning early postpartum follow-up and long-term evolution of children born from acromegalic women, as well as considerations concerning the impact of pregnancy on maternal disease burden. Immediately after birth, placental GH starts to decrease whilst IGF1 increases most probably at higher levels than pre-conception, as well as pituitary GH [38,39]. Magnetic resonance imaging is useful to assess the adenoma expansion, especially in cases with macroadenomas or those without prior neurosurgery [89,90,91,92]. The decision of re-starting DAs will limit breastfeeding. We do not have enough evidence to consider lactation as a risk factor for somatotropinoma increase; thus, lactation seems safe; however, breastfeeding continues to stimulate hyperprolactinemia, which is part of the acromegaly picture [123,127,128,129]. First-generation SSAs are excreted in maternal milk; that is why a certain risk cannot be entirely ruled out, while there are only a few published cases [122,127,128,129,130]. For example, there is one case from 2021 which introduces a 32-year-old female with acromegaly who underwent SSAs during the entire period of pregnancy and 12 months postpartum while lactating. The child was followed for 15 years and had normal growth and development [127]. Another report from 2016 introduces a 25-year-old woman who was admitted after normal delivery and breastfeeding for one year when she experienced amenorrhea post-lactation. The evaluations confirmed acromegaly-related hypogonadism and mild hyperprolactinemia which turned out to be caused by a somatotropinoma with visual field involvement due to suprasellar extension [128]. One study on 16 pregnancies from 6 acromegalic women showed that IQ (intelligence quotient) score and general health status is similar with children with non-acromegalic mothers. However, 2 out of the 16 newborns were large for gestational age as newborns, while another two were identified as tall for their biological age during childhood [129].

We also introduce a new case of acromegalic pregnancy with long-term follow-up. This is a 47-year-old female who was admitted at the age of 34 for suggestive features of acromegaly (and persistent headache), which were confirmed (IGF1 of 1327 ng/mL, with normal levels between 87 and 238 ng/mL) (Figure 2).

She associated hyperprolactinemia (a prolactin value 3 times above the normal upper limit) and mild hypercholesterolemia. A somatotropinoma of 3.3 cm maximum diameter was referred to neurosurgery, but was only partially successfully resected (a post-operatory adenoma of 3.1 cm); thus, she was offered monthly octreotide LAR 30 mg and cabergoline at 2 mg/week (Figure 3).

Ten months after hypophysectomy, she presented for a periodical checkup, but turned out to be 10 weeks pregnant. Both drugs were stopped. Of note, the patient had a history of six successful spontaneous pregnancies before she ever came to our attention. This time, she experienced gestational DM (which required insulin therapy) and HBP, but delivered a healthy newborn (length of 57 cm, weight of 4000 g) by cesarean (week 39). The breastfeeding was resumed due to restarting therapy with SSAs and cabergoline. The boy and the mother were followed for 12 more years. The child had normal growth and development. The female patient, however, had a progressive biochemical disease that required different pharmacological regimes; 5 years postpartum she had an IGF1 of 812 ng/mL (normal: 75–267 ng/mL) under lanreotide Autogel 120 mg/month, cabergoline 2 mg/week and pegvisomant 80 mg/week (of note, she did not tolerate a higher dose of cabergoline and refused radiotherapy, while pasireotide was not available at that time). She was referred for a second neurosurgery (a pre-operatory tumor of 3.8 cm and a post-operatory maximum diameter of 2.8 cm was identified, with invasion into the right cavernous sinus). She continued with the same medical treatment until the present time (another 6 years since her second pituitary resection), the most recent evaluation showing a suboptimal control of the disease, but improved when compared to her medical records—an IGF1 of 365 ng/mL (normal: 83–220 ng/mL) with stationary imaging aspects (Figure 4).

The third aspect we will discuss involves a general look over the published data, gathering original papers we have so far. Over the years, the relatively safe profile of pregnancy in acromegalic subjects remained stationary in most cases. When it comes to particular concerns, the presence of cardio-metabolic complications rather than somatrotropinoma expansion or fetal outcomes seems to matter. We identified three papers that revisited the original publications based on different methods taking into consideration the overall low level of statistical evidence and the fact that restrictive inclusion criteria might limit most of the studies since the approach is heterogamous.

In 2012, a systematic review found 34 prior cases and the authors added a new series of 13 cases (a total of 47 pregnancies) [131]. One article from 2016 mentioned a total of 174 pregnancies [132]. Bandeira BD et al. included 19 studies in 2022, a total of 273 pregnancies (*N* = 211 acromegalic women); acromegaly control was achieved during pregnancy in 62% of cases and somatotropinoma growth was identified in 9% of the entire cohort. No maternal-fetal death was registered. The rate of a more severe glucose profile in persons with pre-pregnancy DM or newly detected gestational DM was 9% and 6% for aggravated HBP and/or pre/eclampsia. The rate of premature labor was of 9%, spontaneous miscarriage 4%, small for gestational age 5% and 1% concerning congenital malformations [88].

We identified a total of 24 original papers that included 13 original studies which, as mentioned, had various profiles of statistical evidence, enrolling between 3 and 141 acromegalic pregnancies per study (or case series), and 11 case reports of a single case, a total of 344 pregnancies and an additional new one (thus, to our knowledge, this is the largest number of pregnancies concerning maternal acromegaly). The original data published within the last decade are observational studies underlying small (mostly) cohorts of acromegalic females or even single cases or pregnancies in patients with different types of pituitary tumors, somatotropinoma being one of them. The studied subgroups were formed either between individuals with controlled disease (under pharmaco-therapy) or suboptimal treated acromegaly; either between subjects who were further exposed to targeted therapy for GH excess or who respected the general recommendations to resume the medication at the moment of conception or pregnancy confirmation; or between first-trimester exposure to SSAs versus longer; and also in patients with GH excess recognition before gestation or during pregnancy; with pre-conception neurosurgery for pituitary adenoma or not [39,51,75,76,77,78,93,94,95,96,97,102,109,111,118,119,120,122,124,125,127,128,129,131] (Table 1).

The forth consideration on the topic concerning maternal acromegaly and pregnancy includes an ideal evolution from acromegaly to pregnancy, meaning that a young patient should receive an early diagnostic of a somatotropinoma before suggestive clinical features, and cardiovascular and metabolic complications occur, followed by prompt intervention in order to cure the patient [133,134]. Sustained remission or cure in the absence of medical therapy is mostly due to successful selective hypophysectomy (rarely, and/or radiotherapy) and, exceptionally, in rare cases with spontaneous or medication-induced pituitary apoplexy (of note, DM, vascular comorbidities and COVID-19 infection might be contributors to this particular type of evolution) [133,135,136,137,138,139].

Another aspect includes the current protocols/guidelines concerning pregnancy in females with different types of pituitary adenomas. The Clinical Practice Guideline from 2021 on behalf of the European Society of Endocrinology addresses pituitary tumours during pregnancy including acromegaly (taking into consideration 159 pregnancies from prior published cases). The recommendation of stopping medical therapy after pregnancy was confirmed (which was identified in 71% of cases) and is not related to a teratogenic risk of either the disease itself or the medication. GH and IGF1 should not be assessed during pregnancy, while the surveillance includes a multidisciplinary team [140]. The decision of performing pituitary imaging is restricted to the cases with tumour growth evidence as seen in other hypophyseal adenomas amid pregnancy; also, the substitution of associated hypopituitarism is similar with these conditions [140].

A sixth aspect includes the first diagnostic of acromegaly during gestation which was reported in two cases [77,96]. One female received the confirmation of acromegaly as a retrospective diagnostic one year after lactation [128]. GH overproduction coming from a somatotropinoma was confirmed in both cases between weeks 11 and 12 of amenorrhea, while the patients experienced symptoms related to large pituitary adenomas (pituitary mass syndrome). Baseline GH and IGF1 assays add no value to the acromegaly confirmation or surveillance amid pregnancy, but suggestive clinical features and pathological report after hypophysectomy (neurosurgery was postponed after delivery in one case) are prone to acromegaly confirmation [77,96,140].

There are no specific studies that address pregnancy ratio and evolution in patients whilst in prolonged remission/cure; they are probably similar to the general population by the same age, unless residual cardio-metabolic complications might complicate the materno-fetal evolution. Further studies on paternal active acromegaly while the partner achieved pregnancy are needed. For the moment, the topic of maternal acromegaly and pregnancy remains in the area of individual-based decisions under the auspices of the evidence-based medicine, which is not very generous for the moment. However, a rather optimistic note concerning the overall safety of pregnancy in these patients should be relevant and the expected expansion of assisted reproduction techniques will widen this information.

## 11. Conclusions

Pregnancy in women with acromegaly seems uneventful and safe, with a similar risk regarding the women of the same age without acromegaly, especially in patients with pre-pregnancy optimal disease control, as reflected by IGF1 levels. The cardio-metabolic profile (mostly, DM and HBP) rather than the evolution of acromegaly/somatotropinoma itself represents the main issue of a successful pregnancy. Specific medication addressing GH-IGF1 excess does not display a teratogenic profile, but it is restricted to cases with headache, visual impairment and/or tumor expansion. The time-frame of pharmaco-intervention should be limited to the first trimester, if not withdrawn at the moment of pregnancy confirmation. Postpartum, SSAs and pegvisomant seem safe; DAs will stop breastfeeding. Due to the limited number of publications, a personalized approach is required involving a complex team of specialists. Further increase of pregnancy cases is expected due to the development of assisted reproduction techniques and better acromegaly management.

## Figures and Tables

**Figure 1 diagnostics-12-02669-f001:**
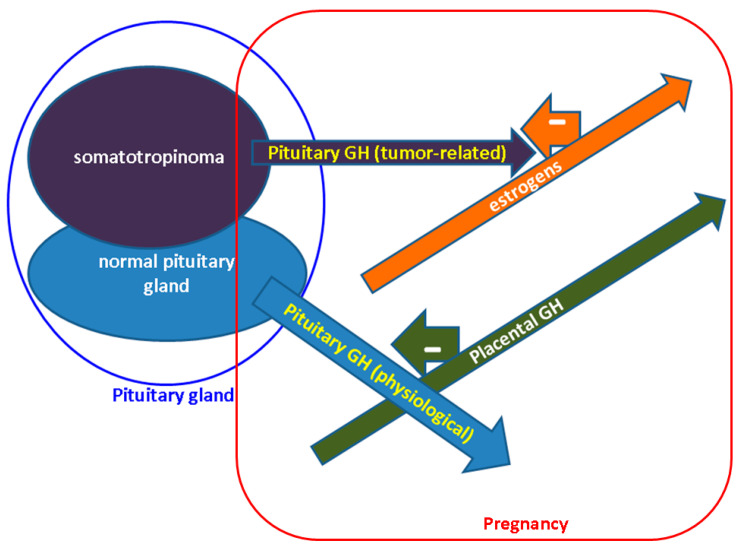
GH amid physiological and acromegalic pregnancy. In non-acromegalic pregnancy pituitary GH is inhibited by the increasing placental GH which takes control of IGF1 levels. In acromeglaic pregnancy, somatotropinoma continues to produce GH, but its actions are antagonized due to estrogen-induced blockade. Abbreviations: GH = Growth Hormone (see references [30,31,32,33,34,35,36,37,38,39,40,41,42,43]).

**Figure 2 diagnostics-12-02669-f002:**
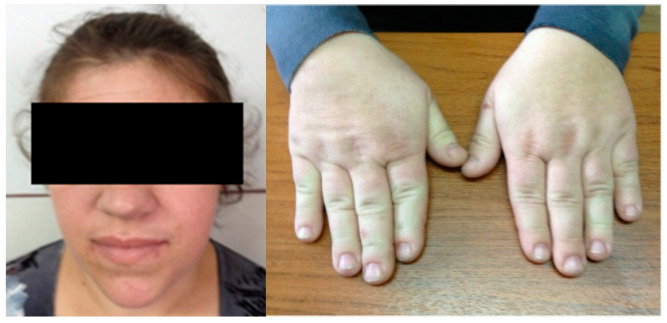
This is a 34-year-old acromegalic patient with suggestive facial (**left**) and hand (**right**) features for GH excess.

**Figure 3 diagnostics-12-02669-f003:**
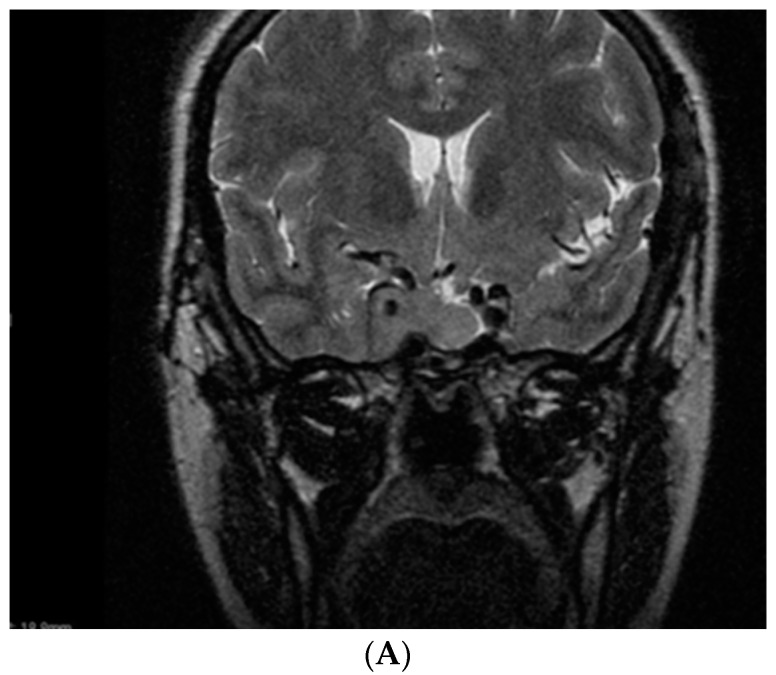
Magnetic resonance imaging of the somatotropinoma (2.4 by 3.3 by 2.5 cm) at initial diagnostic. (**A**) Coronal post-contrast aspect. (**B**) Sagittal post-contrast aspect. (**C**) Post-operatory aspect (2 months since selective trans-sphenoidal hypophysectomy)—a somatotropinoma of 2.4 by 3.1 by 2.2 cm.

**Figure 4 diagnostics-12-02669-f004:**
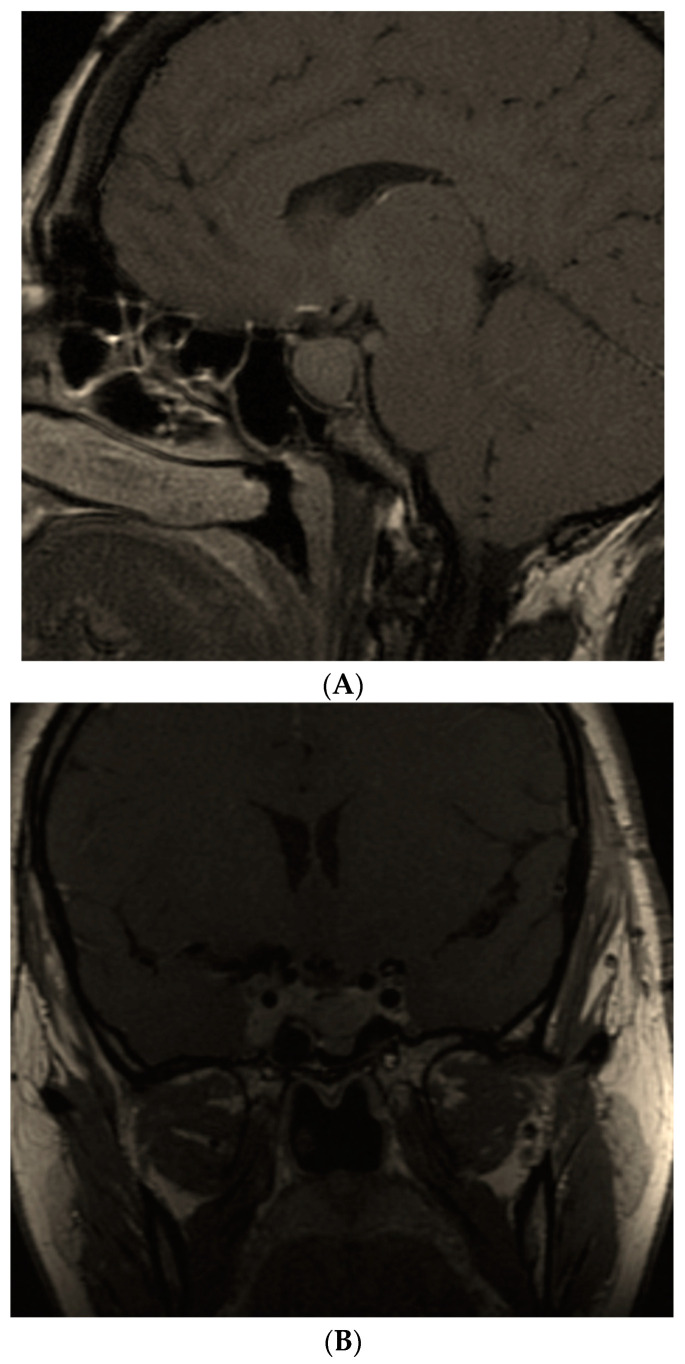
Magnetic resonance imaging of the somatotropinoma during follow-up. (**A**) Before second pituitary surgery (5 years postpartum)—a somatotropinoma of 3.8 by 0.8 by 1.1 cm, with superior and right extension and invasion at the level of cavernous sinus. (**B**) During follow-up: 2 months since second neurosurgery—a pituitary macroadenoma of 2.8 by 0.8 by 1.0 cm with right cavernous sinus invasion. (**C**) The most recent evaluation under triple medical therapy—6 years since the second pituitary intervention (status quo versus 3.B.)

**Table 1 diagnostics-12-02669-t001:** Studies on acromegaly and pregnancy that are published according to our methodology; we only included the studies concerning female acromegalic subjects, not other pituitary tumors; the order of display is based on publication date starting with the most recent (please see references [39,51,75,76,77,78,93,94,95,96,97,102,109,111,118,119,120,122,124,125,127,128,129,131]).

First Author/ Year of Publication/ Reference No.	Type of Study	Studied Population	Results
Das L. 2021 [75]	Retrospective study	*N* = 14 pregnancies from 12 ACM women N1 = 5 active disease N2 = 9 controlled ACM	100% had macroadenomas 2 years: median from hypophysectomy to pregnancy confirmation 13/14 term delivery, normal APGAR 1/14 HBP and preterm birth 0%: congenital malformations N1 = N2: materno-fetal outcomes N1 = N2: postpartum GH
Babinska A. 2021 [127]	Case report	1 female case with ACM	1 healthy new born → followed up for 15 years.
Meoni G. 2020 [95]	Retrospective study	*N* = 141 pregnancies from 127 ACM women N1 = 67 pregnancies in 62 females + SSAs N2 = 74 pregnancies in 65 females treated+ no SSAs	N1 = N2: materno-fetal outcomes Maternal: DM, HBP, headache, delivery mode) Fetal: birth term, height and weight of the new born 1/14 case of congenital malformation (N1)—ureteral stenosis
Guarda FJ. 2020 [111]	Case series	*N* = 4 pregnancies in 3 ACM females (one twin pregnancy)	*N* = 1 female (2 pregnancies)—exposure to PEG until 3 days before embryo transfer *N* = 2 females—exposure to PEG until the moment of pregnancy confirmation No materno-fetal outcomes
Vialon M. 2020 [77]	Retrospective study	*N* = 14 pregnancies in 11 ACM females	*N* = 7 (50%) had gestational DM
Wise-Oringer BK. 2019 [125]	Case report	1 female with X-linked acrogigantism	Prenatal diagnostic of the same disease as the mother → successful pregnancy
Hannon AM. 2019 [96]	Case report	1 female with ACM diagnostic within week 11	Use of octreotide s.c. 100→150 µg/day to control tumor expansion in order to prevent visual loss (she refused surgery)
Hannon AM. 2019 [118]	Irish Pituitary Study	*N* = 17 pregnancies in 12 ACM females	6/17: DA exposure among pregnancy (SSAs was stopped before gestation) Maternal outcomes: no tumor expansion, no visual field event, no case of gestational DM Fetal outcomes: 15/17: healthy newborns at term 1/17: pre-eclampsia—related emergency cesarean (week 32) 1/17: elective C-section for twin pregnancy (week 35)
Dicuonzo F. 2019 [97]	Case report	1 female with giant, inoperable macroadenoma	1 healthy newborn at term
Jallad RS. 2018 [78]	Retrospective study	*N* = 31 pregnancies in 20 ACM females	4/31 abortion 45%: the rate of HBP worsening 32%: the rate of glucose profile worsening 0%: maternal-fetal death 2/27: congenital malformation 1/27: macrosomia
Dogansen SC. 2018 [76]	Retrospective study	*N* = 13 pregnancies in 45 ACM females	7.7%: the rate of gestational DM and HBP 0%: tumor expansion
Karaca Z. 2018 [120]	Retrospective study	*N* = 21 pregnancies in ACM women (a cohort of 113 cases with pituitary tumors)	Congenital malformations: unilateral congenital cataract, craniosynostosis and microcephaly
Tomczyk K. 2017 [122]	Case report	1 female with ACM with pre-conception partial hypophysectomy for macroadenoma	1 healthy newborn at term
Lambert K. 2017 [121]	Prospective study	*N* = 3 pregnancies in ACM women (a cohort of 71 cases with pituitary tumors)	No pregnancies outcome: HBP, DM, preterm labor, stillbirth
Teltayev D. 2917 [102]	Case report	1 female with ACM	1 healthy newborn at term
Căpăþînă C. 2016 [51]	Case report	1 female with ACM with pre-conception debulking hypophysectomy and gammaknife radiotherapy for macroadenoma	1 healthy newborn at term
Viani S. 2016 [124]	Case report	1 female with ACM	subcutaneous implantable defibrillator for secondary prevention of sudden cardiac death
Haliloglu O. 2016 [129]	Longitudinal study	*N* = 16 pregnancies in 6 ACM females	2/16: large for gestational age 2/16: tall for the age during childhood.
Hara T. 2016 [128]	Case report	1 female with ACM	1 healthy newborn (diagnostic of acromegaly after 1 year of lactation)
van der Lely AJ. 2015 [109]	ACROSTUDY	*N* = 35 pregnancies with pegvisomant exposure (27/35 females with ACM)	Data obtained for 10 full-term healthy newborns 5 ACM females: elective abortion 2 ACM females: spontaneous abortion 1 ACM female: ectopic pregnancy
Dias M. 2013 [39]	Prospective study	*N* = 10 pregnancies in 8 ACM females Versus N1 = 64 control pregnancies	*N* = N1: 1 case of HBP/preeclampsia and 1 case of DM
Koshy TG. 2012 [93]	Case report	1 female with ACM	Pituitary surgery (week 22) → cabergoline (1 mg/week) → full-term, healthy new born (cesarean)
Cheng S. 2012 [131]	Case series 343	*N* = 13 pregnancies in ACM females	No teratogenic effect
Kasuki L. 2012 [94]	Case report	1 female with ACM (pre-conception surgery)	1 healthy newborn (week 37, cesarean)

Abbreviations: *N* = number of patients; ACM = acromegaly; DM = diabetes mellitus; HBP = high blood pressure; SGA = small for gestational age; SSA = somatostatin analogues; PEG = pegvisomant; s.c. = subcutaneous.

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
