# Peer review of "Approach of Acromegaly during Pregnancy"

_diagnostics, 2022, doi:10.3390/diagnostics12112669_

Round 1

Reviewer 1 Report

In the present manuscript, Popescu et al. reviewed the clinical picture,  diagnosis and treatment of acromegaly during pregnancy. The review included the papers published during the last 10 years. Functioning or non-functioning pituitary adenoma during pregnancy is a challenging subject and it was extensively discussed in the literature. There are still some controversial issues in the diagnosis and in the treatment of pituitary adenomas including acromegaly during pregnacy. For this reason the present review makes sense. My concerns are as follows;

1- European Society of Endocrinology has recently published a guideline in pituitary adenoma during pregnancy (Luger A et al. Eur J Endocrinol, 2021). Are there differences between that guideline and the current review in terms of the diagnosis, treatment and maternal-fetal outcomes in pregnant women with acromegaly?

2-I am wondering what are the novel findings which were recently reported and mentioned in the current review but have not been reported previosly?

3-In introduction section, they summarized the features of acromegaly including demographic features, clinical manifestations and treatment modalities in acromegalic patients. Since the topic of the review is acromegaly and pregnancy, it would be better to give some brief information regarding physiological changes in the anatomy and hormonal status of pituitary gland during pregnancy (Karaca Z et al. Eur J Endocrinol, 2010) in this section.

4-I recommend a shematic representation summarizing the normal physiological changes in GH-IGF-1 axis and GH-IGF-1 status of acromegaly during pregnancy would be very useful to make the topic more understandable.

5-The reference list should be rearrenged. For example, Karaca Z et al. has been cited as 120th in the paper, but it is cited as 119th on the “References”.

6-The diagnosis of acromegaly during pregnancy is another controversial issue. Do the authors discuss this problem in detail?

Author Response

Response to Reviewer 1 Comments

Dear Reviewer,

Thank you very much for your time and your effort to review our manuscript.

We are very grateful for providing your valuable feedback on the article.

Here is our response and related amendment that has been made in the manuscript according to your review (marked in yellow color).

In the present manuscript, Popescu et al. reviewed the clinical picture, diagnosis and treatment of acromegaly during pregnancy. The review included the papers published during the last 10 years. Functioning or non-functioning pituitary adenoma during pregnancy is a challenging subject and it was extensively discussed in the literature. There are still some controversial issues in the diagnosis and in the treatment of pituitary adenomas including acromegaly during pregnacy. For this reason the present review makes sense.

Thank you very much.

My concerns are as follows; European Society of Endocrinology has recently published a guideline in pituitary adenoma during pregnancy (Luger A et al. Eur J Endocrinol, 2021). Are there differences between that guideline and the current review in terms of the diagnosis, treatment and maternal-fetal outcomes in pregnant women with acromegaly?

The guideline from 2021 does not represent an original study. It takes into consideration 159 pregnancies (a total of 47 papers from a different timeline than our paper which represents the basis of the sections 4.2, followed by sections 4.4. to 5.6.). Our article takes into consideration a larger number of pregnancies based on a different methodology thus some hard data are different. As you suggested, we included a section on Discussion referring to this guideline that, among others, includes acromegaly and pregnancy. As we mentioned in Methods, we did not include data on non-GH- producing tumors. Thank you

I am wondering what are the novel findings which were recently reported and mentioned in the current review but have not been reported previosly?

As mentioned as Methods, this is a narrative review which takes into considerations already published papers, so new hard data are not expected from the start (except from the new case which is probably among the cases with the longest period of surveillance after pregnancy). This type of article is a well-known one, providing practical points from a multidisciplinary perspective upon literature review. However, the article brings together data from different studies with different methods, aims, and approaches, with no restriction regarding the levels of statistical significance. Particular for this topic (acromegaly and pregnancy), almost each of the original studies has a different methodology and, as we you mentioned, there are still controversies and open issues. As we mentioned, too, the major aspect of addressing this topic is the fact that the number of pregnancies in maternal acromegaly is expected to continuously increase due to the recently proved safe profile of medical therapy and a larger access to assisted reproductive technologies. The most important conclusions are highlighted at each section according to the chapters of the article. Thank you

In introduction section, they summarized the features of acromegaly including demographic features, clinical manifestations and treatment modalities in acromegalic patients. Since the topic of the review is acromegaly and pregnancy, it would be better to give some brief information regarding physiological changes in the anatomy and hormonal status of pituitary gland during pregnancy (Karaca Z et al. Eur J Endocrinol, 2010) in this section.

Thank you. Section 3 is dedicated to this aspect. We did not cite papers that were not published within the last decade.

I recommend a shematic representation summarizing the normal physiological changes in GH-IGF-1 axis and GH-IGF-1 status of acromegaly during pregnancy would be very useful to make the topic more understandable.

Thank you. This figure has already been published in European Journal of Endocrinology in 2017 and we respectfully do not intend to reproduce it. https://eje.bioscientifica.com/view/journals/eje/177/1/R1.xml, but we did cite this paper (reference no 123.)

However, based on your suggestion we introduced a new representation of this particular aspect.

The reference list should be rearrenged. For example, Karaca Z et al. has been cited as 120th in the paper, but it is cited as 119th on the “References”.

Thank you. We corrected it.

The diagnosis of acromegaly during pregnancy is another controversial issue. Do the authors discuss this problem in detail?

Thank you.  This is a very interesting point. We introduced a subsection at Discussions.

Thank you very much.

Reviewer 2 Report

Thank you for asking me to provide a review of this article, which has a subject of high interest nowadays, as although not so many, there is an important number of pregnant women who suffer from different endocrinologycal diseases such as acromegaly and materno-fetal outcomes regarding pregnancies in this kind of pathology deserves to be studied more.

   The study was a narrative review on acromegaly that approaches cardio-metabolic features (CMF), somatotropinoma expansion (SE), management adjustments (MA) and maternal and fetal outcomes during pregnancy in women known with acromegaly. 24 original papers were revised in this analysys, which consisted of 13 studies and 11 case reports, gathering 344 pregnancies and 1 unpublished report, which from my point of view is quite sufficient for this kind of study. Also, the papers revised were from January 2012 till September 2022, which is a large period of time and surely were enough to draw conclusions. 

  Unfortunatelly, regarding the structure and accuracy of the phrases, the manuscript has indeed well structured information, with supported evidence and well structured phrases, but there are some English writting problems that need to be corrected so that the paper could be published in well condition. 

  The manuscript is original and well defined and so, the results provide an advance in current knowledge. The results are being interpreted appropriately and are significant, as well as are the conclusions, which are, of course, supported by the results. So the article is written in an appropriate way. 

  The study is correctly designed and the analysis is being performed at high standards, so the data are robust enough to draw the conclusion. 

  Surely the paper will attract a wide readership. 

  The English language is  well understandable, but has some writting mistakes, which can easily be corrected, so that the article could be of highest quality.

  I only have a few things to add in the lines below, strictly regarding the writting techniques, but it is clear that the article is completely adequate and deserves to be published: 

Line 86: „,” before „causing”

Line 90: trans-sphenoidal, not „trans-shenoidal”

Line 90: hypophysectomy, not „hypohysectomy”

Line 107: we revised, not „we revisited”

Line 107: underlying, not „underling”

Line 117: only 1 space between „inducing” and „a”

Line 117: only 1 space between „a” and „pro diabetogenic”

Line 127: by the hypothalamus, not „by hypothalamus”

Line 130: on the fetus, not „on fetus”

Line 266: overall, not „overall all”

Line 391: case series, not „cases series”

Line 591: only 1 space between „.” and „Bandeire”

Line 613: only 1 space between „with” and „pre-conception”

Line 613: only 1 space between „pre-conception” and „neurosurgery”

Line 613: only 1 space between „neurosurgery” and „for”

Line 613: only 1 space between „for” and „pituitary”

Line 613: only 1 space between „pituitary” and „adenoma”

Line 613: only 1 space between „adenoma” and „or”

Line 613: only 1 space between „or” and „not”

Author Response

Response to Reviewer 2 Comments

Dear Reviewer,

Thank you very much for your time and your effort to review our manuscript.

We are very grateful for providing your valuable feedback on the article.

Here is our response and related amendment that has been made in the manuscript according to your review (marked in yellow color).

Thank you for asking me to provide a review of this article, which has a subject of high interest nowadays, as although not so many, there is an important number of pregnant women who suffer from different endocrinologycal diseases such as acromegaly and materno-fetal outcomes regarding pregnancies in this kind of pathology deserves to be studied more.The study was a narrative review on acromegaly that approaches cardio-metabolic features (CMF), somatotropinoma expansion (SE), management adjustments (MA) and maternal and fetal outcomes during pregnancy in women known with acromegaly. 24 original papers were revised in this analysys, which consisted of 13 studies and 11 case reports, gathering 344 pregnancies and 1 unpublished report, which from my point of view is quite sufficient for this kind of study. Also, the papers revised were from January 2012 till September 2022, which is a large period of time and surely were enough to draw conclusions. Unfortunatelly, regarding the structure and accuracy of the phrases, the manuscript has indeed well structured information, with supported evidence and well structured phrases, but there are some English writting problems that need to be corrected so that the paper could be published in well condition. 

Thank you. We revised the paper.

The manuscript is original and well defined and so, the results provide an advance in current knowledge. The results are being interpreted appropriately and are significant, as well as are the conclusions, which are, of course, supported by the results. So the article is written in an appropriate way. The study is correctly designed and the analysis is being performed at high standards, so the data are robust enough to draw the conclusion. Surely the paper will attract a wide readership. The English language is  well understandable, but has some writting mistakes, which can easily be corrected, so that the article could be of highest quality. I only have a few things to add in the lines below, strictly regarding the writting techniques, but it is clear that the article is completely adequate and deserves to be published:  

Line 86: „,” before „causing”

Thank you. We corrected it.

Line 90: trans-sphenoidal, not „trans-shenoidal”

Thank you. We corrected it.

Line 90: hypophysectomy, not „hypohysectomy”

Thank you. We corrected it.

Line 107: we revised, not „we revisited”

Thank you. We corrected it.

Line 107: underlying, not „underling”

Thank you. We corrected it.

Line 117: only 1 space between „inducing” and „a”

Thank you. We corrected it.

Line 117: only 1 space between „a” and „pro diabetogenic”

Thank you. We corrected it.

Line 127: by the hypothalamus, not „by hypothalamus”

Thank you. We corrected it.

Line 130: on the fetus, not „on fetus”

Thank you. We corrected it.

Line 266: overall, not „overall all”

Thank you. We corrected it.

Line 391: case series, not „cases series”

Thank you. We corrected it.

Line 591: only 1 space between „.” and „Bandeire”

Thank you. We corrected it.

Line 613: only 1 space between „with” and „pre-conception”

Thank you. We corrected it.

Line 613: only 1 space between „pre-conception” and „neurosurgery”

Thank you. We corrected it.

Line 613: only 1 space between „neurosurgery” and „for”

Thank you. We corrected it.

Line 613: only 1 space between „for” and „pituitary”

Thank you. We corrected it.

Line 613: only 1 space between „pituitary” and „adenoma”

Thank you. We corrected it.

Line 613: only 1 space between „adenoma” and „or”

Thank you. We corrected it.

Line 613: only 1 space between „or” and „not”

Thank you. We corrected it.

Thank you very much.

Reviewer 3 Report

The authors of this review examined  cardio-metabolic features(CMF), possibloe somatotropinoma expansion(STE), and management adjustment, occurring during pregnancy of females with acromegaly, paying also attention to maternal-fetal outcomes(MFO?. To this purpose they identified 24 papers reviewed on PubMed between January 2012 and September 2022, including 13 studies and 11 single case reports, for a total of 344 pregnancy , both spontaneous and due to therapy for infertility.  They  found that pregnancy in women with acromegaly seems uneventful and safe, with a similar risk  to healthy women of the same age  except for cardiometabolic disorders ( mostly DM and HBP) rather than STE, more frequently  occurring  during pregnancy. The  therapies to reduce GH-IGF1 excess, when needed,,  did not increase teratogenic risk and did not  impair long term growth and development of the newborn. They moreover introduced  a new case with some complex features.

COMMENT

I think this review adds little to the knowledge to this topic. even if the new case described by the Authors is of some interest. The authors considered 24 English-published articles  on PubMed from 2012 to September 2022, even if, by clicking on PubMed with the words Acromegaly and Pregnancy, far more  papers are reviewed, some of them also published  before 2012. Moreover, a review  by Bandeira et al ,published in June  2022 (Acromegaly and Pregnancy: a systematic review and meta-analysis , Pituitary,2022; 25(3):352-362) cited by the Authors with the number  88 in Bibliography  with regards  only to the behavior of the somatotropinoma in these women, actually reported similar results also on other aspects covered by the present review., even if referred to 19 studies and 273 pregnancies.  The new case described by the Authors is of some interest.

Author Response

         Response to Reviewer 3 Comments

Dear Reviewer,

Thank you very much for your time and your effort to review our manuscript.

We are very grateful for providing your valuable feedback on the article.

Here is our response according to your review.

The authors of this review examined  cardio-metabolic features(CMF), possibloe somatotropinoma expansion(STE), and management adjustment, occurring during pregnancy of females with acromegaly, paying also attention to maternal-fetal outcomes(MFO?. To this purpose they identified 24 papers reviewed on PubMed between January 2012 and September 2022, including 13 studies and 11 single case reports, for a total of 344 pregnancy , both spontaneous and due to therapy for infertility.  They  found that pregnancy in women with acromegaly seems uneventful and safe, with a similar risk  to healthy women of the same age  except for cardiometabolic disorders ( mostly DM and HBP) rather than STE, more frequently  occurring  during pregnancy. The  therapies to reduce GH-IGF1 excess, when needed,,  did not increase teratogenic risk and did not  impair long term growth and development of the newborn. They moreover introduced  a new case with some complex features. COMMENT. I think this review adds little to the knowledge to this topic. even if the new case described by the Authors is of some interest. The authors considered 24 English-published articles  on PubMed from 2012 to September 2022, even if, by clicking on PubMed with the words Acromegaly and Pregnancy, far more  papers are reviewed, some of them also published  before 2012. Moreover, a review  by Bandeira et al ,published in June  2022 (Acromegaly and Pregnancy: a systematic review and meta-analysis , Pituitary,2022; 25(3):352-362) cited by the Authors with the number  88 in Bibliography  with regards  only to the behavior of the somatotropinoma in these women, actually reported similar results also on other aspects covered by the present review., even if referred to 19 studies and 273 pregnancies.  The new case described by the Authors is of some interest.

Thank you. In addition to the new case, as you mentioned, we respectfully consider that the importance of this article includes several aspects like:

  • This is the largest study on published cases concerning pregnancies in maternal acromegaly
  • We aimed to address only the most recent (last decade) data
  • Specific published data on pregnancy in acromegalic women covers different levels of statistical significance and we did not restrict this aspect
  • We aimed to provide practical data beyond a strictly endocrine perspective (like cardiologic and metabolic aspects, as well as obstetrical issues)
  • We addressed other reviews papers at Discussion to highlight different methods that have been used over the years by applying various methods thus the heterogeneity of these data
  • The subject still has open issues, as we mentioned, acromegaly (not even an active disease) does not limit assisted reproductive techniques that is why the number of such cases is expected to grow.
  • Most of the published papers aim to highlight different aspects of this complex topic, and we observed that particularly on this topic of maternal acromegaly, many papers introduce new cases or series or even studies in addition to a review of the already published data. For instance, reference number 110 or number 131:

Preconception use of pegvisomant alone or as combination therapy for acromegaly: a case series and review of the literature.

Guarda FJ, Gong W, Ghajar A, Guitelman M, Nachtigall LB.Pituitary. 2020 Oct;23(5):498-506. doi: 10.1007/s11102-020-01050-2.

Pregnancy in acromegaly: experience from two referral centers and systematic review of the literature.

Cheng S, Grasso L, Martinez-Orozco JA, Al-Agha R, Pivonello R, Colao A, Ezzat S.Clin Endocrinol (Oxf). 2012 Feb;76(2):264-71. doi: 10.1111/j.1365-2265.2011.04180.x.

Round 2

Reviewer 2 Report

Thank you for requesting to provide a review of this revised article, which has a subject of high interest.

   Regarding the structure and accuracy of the phrases, after the corrections being made, the manuscript has well structured information, with supported evidence and well structured phrases.

  The manuscript is original and well defined. 

  The results provide an advance in current knowledge.       

  The results are being interpreted appropriately and are significant, as well as are the conclusions, which are, of course, supported by the results, so the article is written in an appropriate way. 

  The study is correctly designed and the analysis is being performed at high standards.

  The data are robust enough to draw the conclusion. 

  Surely the paper will attract a wide readership. 

  The English language is  well understandable.

  It is clear that the article is completely adequate and should  be published.

Reviewer 3 Report

I find the Authors' reply satisfactory. The revised paper has been improved also by adding and commenting some new pertinent citations and therefore deserves to be reconsidered for publication on Diagnostics